# A Multi-Agent Reinforcement Learning Framework for Evaluating the U.S. 'Ending the HIV Epidemic' Plan

## Abstract

Human immunodeficiency virus (HIV) is a major public health concern in the United States, with about 1.2 million people living with HIV and 35,000 newly infected each year. There are considerable geographical disparities in HIV burden and care access across the U.S. The 2019 'Ending the HIV Epidemic (EHE)' initiative aims to reduce new infections by 90% by 2030, by improving coverage of diagnoses, treatment, and prevention interventions and prioritizing jurisdictions with high HIV prevalence. Identifying optimal scale-up of intervention combinations will help inform resource allocation. Existing HIV decision analytic models either evaluate specific cities or the overall national population, thus overlooking jurisdictional interactions or differences. In this paper, we propose a multi-agent reinforcement learning (MARL) model, that enables jurisdiction-specific decision analyses but in an environment with cross-jurisdictional epidemiological interactions. In experimental analyses, conducted on jurisdictions within California and Florida, optimal policies from MARL were significantly different than those generated from single-agent RL, highlighting the influence of jurisdictional variations and interactions. By using comprehensive modeling of HIV and formulations of state space, action space, and reward functions, this work helps demonstrate the strengths and applicability of MARL for informing public health policies, and provides a framework for expanding to the national-level to inform the EHE.

## 1 Introduction

Human immunodeficiency virus (HIV) poses a significant public health challenge in the United States (U.S.), with more than 1.2 million individuals living with HIV. In 2019, there were 34,800 new infections reported, equating to a rate of 12.6 infections per 100,000 people HIV.gov (d). The primary treatment for HIV is antiretroviral therapy (ART), which involves the daily use of a combination of HIV medications, referred to as an HIV regimen, to manage the infection. While ART cannot cure HIV, it plays a crucial role in extending the lifespan of people living with HIV (PWH) and preventing the risk of HIV transmission. The purpose of ART is to achieve viral load suppression (VLS), defined as having fewer than 200 copies of HIV per milliliter of blood, making it undetectable through viral load testing HIV.gov (b). Additionally, pre-exposure prophylaxis (PrEP) is a medication used by individuals at risk for HIV to prevent infection through sexual contact or injection drug use. PrEP effectively prevents HIV from establishing itself and spreading within the body HIV.gov (a).

Despite the availability of these preventive and treatment measures, reducing the incidence of new HIV infections remains a formidable challenge nationwide. Disparities in infection rates persist across different geographical areas and risk groups. Specifically, 57 jurisdictions, comprising counties and states, accounted for over 50% of all new HIV diagnoses reported in 2016 and 2017 HIV.gov (c). In response to this alarming situation, the U.S. Department of Health and Human Services launched the 'Ending the HIV Epidemic (EHE)' initiative in 2019, with the ambitious goal of eradicating the HIV epidemic in the U.S. by 2030. The EHE initiative aims to achieve two primary objectives: (i) reduce new HIV incidence by 75% in these EHE-designated jurisdictions by 2025 and (ii) decrease overall HIV incidence by 90% across all jurisdictions by 2030 HIV.gov (c). Additionally, the initiative targets achieving the 95-95-95 care continuum benchmarks, which entail ensuring that 95% of PWH are aware of their infections, 95% of those aware of their status are linked to care, and 95% of individuals in care attain VLS. Furthermore, the initiative aims to attain 50% PrEP coverage among eligible individuals by 2025 in EHE jurisdictions and by 2030 nationwide AHEAD (a).

To accomplish these ambitious targets, several essential actions have been identified: (i) prompt diagnosis (testing) of all individuals with HIV, (ii) rapid and effective treatment with ART to achieve VLS, and (iii) adequate prevention measures to reduce new HIV transmissions, including the widespread use of PrEP among high-risk groups. Given the limited resources, finding jurisdiction-specific optimal combinations of these actions, to minimize new HIV infections, would be key to optimal resource allocation.

There are multiple models in the HIV literature that have evaluated the EHE, and can be categorized into two types: independent jurisdictional models, i.e., decisions for each jurisdiction are evaluated individually, or national models, i.e., decisions are evaluated at the aggregated national level, as follows. Krebs et al. (2019) and Zang et al. (2020) developed a framework to construct and calibrate a dynamic compartmental HIV transmission model exclusively for these six U.S. cities (Atlanta (GA), Baltimore (MD), Los Angeles (CA), Miami (FL), New York City (NY), and Seattle (WA), which together account for 24.1% of the total population living with HIV in the U.S.). Using this model Nosyk et al. (2019) projected HIV incidence trajectories for a decade (2020-2030) within these cities, highlighting the need for new strategies to achieve the EHE goal. Nosyk et al. (2020b) identified the optimal combination of strategies that offer both health benefits and cost-effectiveness within these six cities. Focusing on specific risk groups, Krebs et al. (2020) examined the cost-effectiveness of optimal strategies for persons who inject drugs (PWID) within the same six cities. Furthermore, Nosyk et al. (2020a) delved into the impact of racial and ethnic disparities in healthcare access on progress towards the EHE goal within these cities. Expanding beyond these cities, Fojo et al. (2021) employed a combination of HIV testing and PrEP coverage as intervention strategies to project HIV incidence in 32 priority metropolitan statistical areas. Additionally, Wheatley et al. (2022) assessed the cost-effectiveness of PrEP in reducing overall HIV incidence specifically in Atlanta, GA. Among national-level models, Khatami & Gopalappa (2021) explored the optimal combination of HIV testing and retention-in-care strategies towards the EHE goal, examined at the national level. Khurana et al. (2018) investigated the impact of PrEP on overall HIV incidence in the U.S. from 2016 to 2030. Recently, Khatami & Gopalappa (2021) introduced a national HIV elimination model. This model identified an optimal sequence of HIV-testing and retention-in-care rate combinations at 5-year intervals from 2015 to 2070, employing Q-learning, a classical reinforcement learning (RL) algorithm.

The gaps in the above models are as follows. The national models evaluate aggregated national decisions, and thus ignore heterogeneity across jurisdictions. The independent jurisdictional models, though evaluate decisions specific to the jurisdiction, ignore epidemic interactions across jurisdictions, occurring through sexual partnership mixing. Data from national survey and surveillance systems indicate that about 47-65% of sexual partnerships occur between persons in the same county, and the remaining are between persons who reside in different counties. Tatapudi & Gopalappa (2022) To address these gaps, recently, Tatapudi & Gopalappa (2022) developed a national simulation model comprised of 96 *interacting* sub-jurisdictions. This is ideal to evaluate jurisdiction-specific decisions but in a national context by considering the impact of jurisdictional interactions (from partnership mixing and decisions). However, there does not currently exist a model that has evaluated suitable solution algorithms. Considering the complexity of the disease dynamics and time-dependent decision-making problem, deep reinforcement learning (DRL) offers a promising approach for deriving (near-)optimal policies. However, considering the dimensionality of the problem, modeling this as a single-agent RL, with the state space and action space a conjunction of all jurisdictions, would make it computationally challenging to solve.

We formulate this problem as a multi-agent reinforcement learning (MARL) problem. Each jurisdiction is an agent, who makes independent decisions in an environment with interactions across jurisdictions. An agent's objective is to identify an optimal policy, i.e., optimal combinations of testing, retention-in-care, and prevention rates, for every year over the period 2020 to 2030, specific to its jurisdiction, to eliminate HIV. Though the decisions are decentralized, there is cooperation between agents, i.e., the state of the system at any time-step is known to all agents and all agents have a common objective to eliminate HIV, which represents reality as CDC National HIV Surveillance Systems monitor and publish the state of the epidemic for every jurisdiction. However, at each decision-making step (every year), as each agent is evaluating its optimal policy, and all agents are simultaneously doing so, the state transitions and rewards are influenced by the actions take by all agents, thus adding non-stationary to the state-transition, making this suitable for the application of MARL.

To evaluate the advantage of MARL, we compare the results with two other approaches, a national aggregated single-agent RL (A-SARL), and an independent jurisdiction single-agent RL (I-SARL). A-SARL and I-SARL are representative of the current two types of models in the HIV literature, discussed above. I-SARL representative of national aggregated model, and the A-SARL representative of independent jurisdictional models. As demonstration, we conduct a numerical study on a smaller scale, focusing on the jurisdictions in two states, California and Florida.

The primary contribution of this work is the development of a MARL framework for evaluation of a public health national strategic plan such as the EHE, to identify optimal jurisdiction-specific decisions in the context of jurisdictional interactions. Another noteworthy contribution is the meticulous formulation of the key components modeling of the underlying Markov decision process (MDP). Specifically, the state within the MDP formulation encapsulates all necessary information in a concise format for decision-making in each jurisdiction. The reward signal is finely tuned using the new infection rate and the available budget for HIV care and treatment in each jurisdiction. Through our experimental findings, we demonstrate the effectiveness of our sequential decision-making framework for generating intervention policies across varying jurisdictional budgets.

The rest of the paper is organized as follows: Section 2 discusses related literature. Section 3 offers an overview of our simulation model and proposed approaches. Section 4 presents computational results and associated discussions. Lastly, Section 5 provides the paper's conclusion.

## 2  Literature Review

In addition to the recent literature studies on the HIV public health challenge mentioned in Section 1, several other relevant studies deserve attention. Kok et al. (2015) developed an optimal resource allocation model using a system dynamics approach to the continuum of HIV care, aiming to minimize new HIV infections in Vancouver. Okosun et al. (2013) conducted an analysis to determine the conditions for optimal control of the disease, considering factors such as the effective use of condoms, treatment regimes, and screening of infectives. Gopalappa et al. (2012) and Lin et al. (2016) focused on assessing the cost-effectiveness of various interventions in HIV prevention. Additionally, several studies Lasry et al. (2011), Zaric & Brandeau (2001), Lasry et al. (2012), Yaylali et al. (2016), Juusola & Brandeau (2016), Gromov et al. (2018), Yaylali et al. (2018) developed optimization models for allocating HIV funds to enhance the effectiveness of HIV prevention efforts. In the context of another recent public health challenge, COVID-19, Matrajt et al. (2021), Yu et al. (2021), Han et al. (2021) utilized optimization models to determine vaccine prioritization strategies when vaccine supply is limited. Libotte et al. (2020) and Tsay et al. (2020) investigated optimal control strategies for managing the COVID-19 pandemic. Tang et al. (2022) formulated a bi-objective optimization problem for multi-period COVID-19 vaccination planning, aiming to minimize operational costs and the distance traveled by vaccine recipients. Finally, Rawson et al. (2020) employed an optimization approach to explore efficient strategies for ending COVID-19 lockdowns without overwhelming healthcare services.

Development and application of RL solution methodologies in the public health domain has experienced significant growth in recent times. For instance, Kwak et al. (2021) applied a DRL approach to determine optimal intervention strategies during the COVID-19 pandemic. This included identifying the optimal timing and intensity of lockdowns and travel restrictions for various countries and territories. Libin et al. (2021) employed DRL to model the spread of pandemic influenza in Great Britain, utilizing 379 patches to capture the infection process and develop mitigation policies. Awasthi et al. (2022) proposed a DRL model with a contextual bandits approach to optimize the distribution of COVID-19 vaccines across five different states in India. Furthermore, Allioui et al. (2022) introduced a DRL-based method for extracting masks from COVID-19 computed tomography (CT) images, enabling accurate clinical diagnoses by extracting visual features of COVID-19 infected areas. Jalali et al. (2021) combined RL with optimized convolutional neural networks (CNNs) to reduce ensemble classifier size and enhance model performance. Kompella et al. (2020) developed an agent-based pandemic simulator for COVID-19, employing DRL-based methodology to optimize mitigation policies. Kumar et al. (2021) proposed a model incorporating recurrent neural networks (RNNs) for forecasting COVID-19 cases and integrated DRL to optimize these values based on symptoms. Lastly, Bednarski et al. (2021) investigated the use of RL models to facilitate the redistribution of medical equipment, enhancing the public health response in preparation for future crises.

The primary objective of an RL agent is to make decisions that assess the impact of all actions in terms of short-term and long-term utility for stakeholders. This characteristic makes RL an ideal approach for addressing complex decision problems within public health, such as resource allocation during a pandemic, monitoring and testing strategies, and adaptive sampling for hidden populations Weltz et al. (2022). Several DRL algorithms, including Deep Q-Network (DQN) Mnih et al. (2013), Dueling Double Deep Q-Network (D3QN) Wang et al. (2016), Soft Actor-Critic (SAC) Haarnoja et al. (2018), Proximal Policy Optimization (PPO) Schulman et al. (2017), have been applied to various public health problems Kwak et al. (2021) Kompella et al. (2020) Libin et al. (2021). Notably, PPO has demonstrated strong performance in addressing problems with high-dimensional state spaces and continuous action spaces.

MARL is a subfield of RL where multiple agents interact in a shared environment, and each agent's actions not only affect its own rewards but also impact the rewards and states of other agents. MARL focuses on the study of how agents should learn to make decisions in such interactive and often competitive or cooperative environments. MARL algorithms find applications across various fields, including traffic control, resource allocation, robot path planning, production systems, and maintenance management Oroojlooy & Hajinezhad (2022).Chu et al. (2019) employed a multi-agent advantage actor-critic approach for adaptive traffic signal control within complex urban traffic networks. Their strategy involved forming a cooperative MARL system, with each intersection serving as a local RL agent equipped with its own local observations and partial knowledge of the overall system. Yang et al. (2020) integrated traffic signals and vehicle speeds to enhance urban traffic control performance. They utilized a multi-agent modified PPO method, making adjustments to the clip hyperparameter for improved learning. In the realm of wireless networks, Nasir & Guo (2019) developed a distributively executed dynamic power allocation model using multi-agent deep Q-learning. For large-scale fleet management challenges, Lin et al. (2018) applied contextual deep Q-learning and contextual multi-agent actor-critic algorithms. They treated each available vehicle as an agent and grouped vehicles in the same region at the same time as identical agents. Furthermore, Yu et al. (2020) employed a multi-agent actor-attention-critic mechanism to minimize heating, ventilation, and air conditioning costs in a multi-zone commercial building under dynamic pricing conditions. Their approach took into consideration random zone occupancy, thermal comfort, and indoor air quality comfort.

In this paper, we use a DRL approach to develop a policy-based methodology for optimizing actions in the dynamic and stochastic HIV disease environment. We treat individual jurisdictions as independent agents in a MARL framework to obtain a decentralized decision-making strategy to get closer to achieving the EHE targets, overcoming national model limitations, and accommodating diverse jurisdictional needs and constraints.

## 3 METHODOLOGY

The model framework comprises a compartmental simulation model for simulating disease progression, an MDP tailored for addressing the problem, and a DRL agent tasked with solving the resulting MDP. In this approach, we represent each jurisdiction as an individual agent and employ MARL to facilitate interaction among these agents within the environment. This enables each agent to learn its optimal policy while taking into account the dynamic changes in the environment caused by other agents operating within the same context. Figure 1 provides a schematic overview of our MARL framework where we use compartmental simulation model used in Tatapudi & Gopalappa (2022) as the simulator.

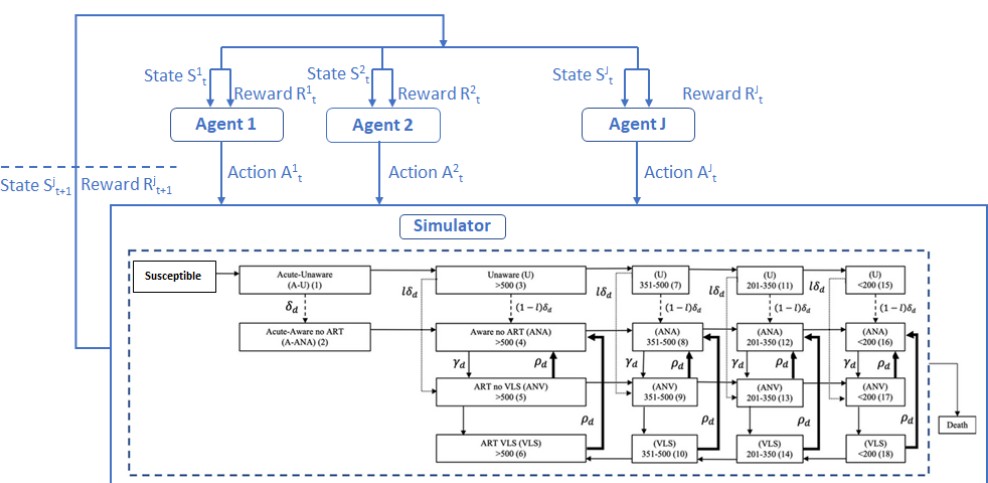

Figure 1: Schematic of the multi-agent reinforcement learning (MARL) framework with compartmental simulation from Tatapudi & Gopalappa (2022).

## 3.1 Simulation Model

The disease progression of HIV is tracked through the compartmental simulation model developed in Tatapudi & Gopalappa (2022). An overview of this compartmental simulation is provided in the simulator of Figure 1. The model encompasses four stages within the care continuum: Unaware, Aware no ART, ART no VLS, and ART VLS. Each of these care continuum stages is further divided into five disease progression stages based on CD4 count: Acute, CD4 > 500, CD4 351-500, CD4 201-350, and CD4 < 200. Additionally, the model includes compartments for individuals who are susceptible to infection and those who have succumbed to the disease, resulting in a total of 22 compartments. In the model, $\delta_d$ represents the diagnostic rate, $\rho_d$ represents the dropout rate, and $\gamma_d$ represents the rate of entry to care in disease stage $d$. Furthermore, the parameter $l$ signifies the proportion of individuals who are linked to care within three months of diagnosis. we model three transmission risk groups, heterosexual males ($HM$), heterosexual females ($HF$), and men who have sex with men ($MSM$). Table A2 presents data for partnership mixing across jurisdictions for each transmission group, and was adopted from Tatapudi & Gopalappa (2022).

## 3.2 Markov Decision Process (MDP)

The MDP can be defined as a tuple $N, S, \mathbf{a}, P, R, \gamma$. Here, $N$ represents the number of agents, where $N = 1$ corresponds to a single-agent MDP. $S$ is the set of environmental states, denoted as $o^j$ for each agent $j$ to represent their local observations. For each agent $j$, $\mathbf{a}^j$ signifies their set of possible actions. Consequently, the total collection of actions, denoted as $\mathbf{a}$, is defined as the Cartesian product: $\mathbf{a} = \mathbf{a}^1 \times \mathbf{a}^2 \times \ldots \times \mathbf{a}^N$. The state transition function, $P$, is a mapping from the current state $s \in S$ and the joint action $\mathbf{a}$ to the next state $s' \in S$ at time $t + 1$. This is represented as $P : S \times \mathbf{a} \to \Delta(S)$, where $\Delta(S)$ denotes the probability distribution over states. The set of reward functions, $R$, includes individual reward functions $R^j : S \times \mathbf{a} \times S \to R$, where $R^j$ signifies the reward for agent $j$ when transitioning from state $(s, \mathbf{a})$ at time $t$ to the next state $s'$ at time $t + 1$. Lastly, the discount factor $\gamma \in [0, 1]$ is utilized to adjust for future rewards. It plays a crucial role in the MDP formulation by determining the weightage of future rewards relative to immediate rewards. The key elements of the MDP formulation are further described below.

### 3.2.1 State

State represents the information about a jurisdiction at a particular time. The state of a jurisdiction $j \in J$ at time $t$ can be represented as,

$$s_t^j = [p_{k,j,t}, \mu_{u,k,j,t}, \mu_{a,k,j,t}, \mu_{ART,k,j,t}, \mu_{prep,k,j,t}; \forall k \in \{HM, HF, MSM\}] \tag{1}$$

where, $p_{k,j,t}$ denotes the proportion of people living with HIV (PWH), diagnosed or undiagnosed, $\mu_{u,k,j,t}$ denotes the proportion of PWH unaware of their infection, $\mu_{a,k,j,t}$ denotes the proportion of PWH aware of their infection but not on ART, $\mu_{ART,k,j,t}$ denotes the proportion of PWH aware of their infection and on ART or on VLS, and $\mu_{prep,k,j,t}$ denotes the proportion of people who have been prescribed PrEP among the people with PrEP indicators, for each risk group $k \in \{HM, HF, MSM\}$ in jurisdiction $j \in J$ at time $t$.

### 3.2.2 Action

The action $\mathbf{a}_t^j$ for agent $j \in J$ at time $t$ represents a combination of three sets of measures: testing, retention-in-care and treatment, and prevention, aimed at controlling HIV infections. These actions are characterized by the changes in the following proportions: (i) the proportion of PWH who are unaware of their infection ($\mu_{u,k,j,t}$), (ii) the proportion of PWH who are aware of their condition and are on ART ($\mu_{ART,k,j,t}$), and (iii) the proportion of individuals taking PrEP as a preventive measure ($\mu_{prep,k,j,t}$). Note that, instead of directly formulating testing and retention-in-care using corresponding rates, we formulated as above because of its attractive mathematical properties that help efficiently constrain the number of action choices and thus improve the chance of convergence of an RL algorithm, as shown in Khatami & Gopalappa (2021).

$$\mathbf{a}_t^j = [a_{unaware,k,t}^j, a_{ART,k,t}^j, a_{prep,k,t}^j; \forall k \in \{HM, HF, MSM\}] \tag{2}$$

where, $a_{unaware,k,t}^j$ denotes the proportion decrease in $\mu_{u,k,j,t}$, $a_{ART,k,t}^j$ denotes the proportion increase in $\mu_{ART,k,j,t}$, and $a_{prep,k,t}^j$ denotes the proportion increase in $\mu_{prep,k,j,t}$, for each risk group $k \in \{HM, HF, MSM\}$ in jurisdiction $j \in J$ at time $t$.

### 3.2.3 STATE TRANSITION FUNCTION

The state transition function calculates the probabilities of a system transitioning from state $s_t$ to the next state $s_{t+1}$ when an action $\mathbf{a}_t$ is taken. For this problem, the state transition probabilities are not available, hence we use the simulator to provide the Markov jumps from one state to another under the action taken by the agent(s).

### 3.2.4 REWARD

Rewards reflect the impact of an action on the current state. Immediate rewards are computed as the negative of the total new infections, as our objective is to minimize this value over time. Notably, each jurisdiction, $j$, has a budget, $B^j$, allocated towards reducing HIV infections RWHAP. In addition to the reward of lowering the infection count, the reward function also contains a penalty term, which is the difference in the cost allocated by the agent and the budget allocated for that jurisdiction. The rewards $R^j_{a^j_t}(s^j_t, s^j_{t+1})$ for an agent $j \in J$ transitioning from state $s^j_t$ to the next state $s^j_{t+1}$ under action $\mathbf{a}^j_t$ can be estimated as follows:

$$R^j_t = -\sum_k i_{k,j,t} - P^j_t \tag{3}$$

where, $i_{k,j,t}$ represents the new infections for risk groups $k \in \{HM, HF, MSM\}$ in jurisdiction $j$ at time $t$. $P^j_t$ is the penalty term, which is calculated as, $P^j_t = C^j_t - B^j_t$, the difference between the cost allocated by agent $C^j_t$ and the budget $B^j_t$ for jurisdiction $j$ at time $t$. Notably, $C^j_t$ is obtained using HIV related cost functions in Khatami & Gopalappa (2021) and estimation of those costs from the simulation model, and the budget $B^j_t$ can be obtained from Ryan White HIV/AIDS Funding Program HRSA.

## 3.3 DEEP REINFORCEMENT LEARNING (DRL) SOLUTION METHODOLOGY

The optimal values for testing, treating, and PrEP can be obtained using the DRL solution methodology. In our MARL framework, each agent samples an action and interacts with the environment, represented by the simulator. Although each agent acts independently, their simultaneous interactions within the same environment can influence the actions taken by all of them. Consequently, each agent receives a reward from the simulator and transitions to the next state.

### 3.3.1 DRL ALGORITHM

Among the various DRL algorithms, PPO Schulman et al. (2017) is known for its stability and efficiency. It is an on-policy algorithm that can be used for environments with continuous action spaces and very large state spaces. Our research problem has the characteristics that are well suited for such an approach. PPO adopts the concepts from trust region policy optimization (TRPO) Schulman et al. (2015). The objective in TRPO, called surrogate advantage, is calculated as:

$$L^{CPI}(\theta) = \hat{E}_t \left[ \frac{\pi_\theta(a_t|s_t)}{\pi_{\theta_l}(a_t|s_t)} \hat{A}_t \right] = \hat{E}_t[r_t(\theta)\hat{A}_t] \tag{4}$$

The probability ratio $r_t(\theta)$ of two policies give a measure of how far the new policy $\pi_\theta$ is from the old policy $\pi_{\theta_k}$. As having a large policy update at a time would destabilize the training, PPO-clip puts a limit on the extent of policy update with the following objective.

$$L^{CLIP}(\theta) = \hat{E}_t \left[ \min(r_t(\theta)\hat{A}_t, clip(r_t(\theta), 1 - \epsilon, 1 + \epsilon)\hat{A}_t) \right], \tag{5}$$

where $\epsilon$ is the hyperparameter that clips the probability ratio and prevents $r_t$ from moving outside the interval $[1 - \epsilon, 1 + \epsilon]$. The probability ratio is clipped with $1 + \epsilon$ or $1 - \epsilon$ depending on whether the advantage is positive or negative. The surrogate objective for PPO-clip can be generalized with the following equation:

$$L^{CLIP}_{(s,a,\theta_l,\theta)} = \min \left( \frac{\pi_\theta(a|s)}{\pi_{\theta_l}(a|s)} A^{\pi_{\theta_l}}(s,a), g(\epsilon, A^{\pi_{\theta_l}}(s,a)) \right) \tag{6}$$

where,

$$g(\epsilon, A) = \begin{cases} (1 + \epsilon)A & A \geq 0 \\ (1 - \epsilon)A & A < 0 \end{cases} \tag{7}$$

**Algorithm 1** MARL Framework for Optimal Interventions in the HIV Epidemic

---

Input: Initialize policy parameters $\theta_l^j$, initialize value function parameters $\phi_l^j$ for $j = 1, 2, ..., N$ agents, set episode length $T$ (here we used $T = 12$ years corresponding to year 2019 to 2030), set number of episodes $M$, set buffer size $B$ (here we used $B = 10$ episodes), set total timesteps $X = 0$, set $l = 0$.

1: Initialize replay buffer $\mathcal{D}_l^j$ for each agent $j$.
2: **for** episode $m = 0, 1, 2, ..., M$ **do**
3:     **for** year $t = 1, 2, ..., T$ **do**
4:         At $t = 1$, extract the initial state for each agent $j$ as:
$$s_t^j = [p_{k,j,t}, \mu_{u,k,j,t}, \mu_{a,k,j,t}, \mu_{ART,k,j,t}, \mu_{prep,k,j,t}; \forall k \in \{HM, HF, MSM\}].$$
5:         Sample actions $\mathbf{a}_t^j$ comprising the change in proportions from each agent $j$ by running policy $\pi_l^j = \pi_{\theta_l^j}$
$$\mathbf{a}_t^j = [a_{unaware,k,t}^j, a_{ART,k,t}^j, a_{prep,k,t}^j; \forall k \in \{HM, HF, MSM\}].$$
6:         Calculate the diagnostic and dropout rates using the change in unaware and ART proportions, respectively.
7:         Simulate the actions in the compartmental simulation model.
8:         Observe next state $s_{t+1}^j$ and reward $R_{t+1}^j$, and collect set of trajectories $\{\tau_t^j\}$, where
$$\tau_t^j = (s_t^j, a_t^j, R_{t+1}^j, done).$$
9:         Add the trajectories to the replay buffers $\mathcal{D}_l^j$.
10:        $X \leftarrow X + 1$
11:        **if** ($X$ **mod** $B \times T$) **= 0 then**
12:            Compute advantage estimates, $A_t^j$ based on the current value function $V_{\phi_l^j}^j$.
13:            Update the policy by maximizing objective (via stochastic gradient ascent):
$$\theta_{l+1}^j = \arg\max_{\theta^j} \frac{1}{|\mathcal{D}_l^j|T} \sum_{\tau_t^j \in \mathcal{D}_l^j} \sum_{t=0}^{T} \min\left( \frac{\pi_{\theta^j}(a_t^j|s_t^j)}{\pi_{\theta_l^j}(a_t^j|s_t^j)} A^{\pi_{\theta_l^j}}(s_t^j, a_t^j), g(\epsilon, A^{\pi_{\theta_l^j}}(s_t^j, a_t^j)) \right).$$
14:            Fit value function by regression on mean-squared error:
$$\phi_{l+1}^j = \arg\min_{\phi^j} \frac{1}{|\mathcal{D}_l^j|T} \sum_{\tau_t^j \in \mathcal{D}_l^j} \sum_{t=0}^{T} \left( V_{\phi_l^j}^j(s_t) - R_t^j \right)^2.$$
15:            Clear replay buffer for each agent $j$.
16:            $l \leftarrow l + 1$
17:        **end if**
18:     **end for**
19: **end for**

---

## 4 EXPERIMENTS AND RESULTS

### 4.1 EXPERIMENTAL SETUP

We adopted the initial data for HIV infection in all jurisdictions for the year 2018 from Tatapudi & Gopalappa (2022). This data encompasses all age groups from 13 to 100 and includes all 22 care continuum stages for three risk groups. The compartmental simulation model, used as the simulator in our study to determine transitions from time $t$ to $t + 1$, was also sourced from the same paper. In our experiments, we selected two distinct groups of jurisdictions, one from California and the other from Florida. Each of these groups consists of eight jurisdictions corresponding to their respective states.

MARL: The formulations were discussed for a MARL setup. In MARL, each jurisdiction within a state is an agent, evaluating an optimal policy for its jurisdiction. At each decision-making step (every year), as each agent is evaluating its optimal policy, and all agents are simultaneously doing so, the state transitions and rewards are influenced by the actions take by all agents.

A-SARL: In the aggregated-SARL, we have one agent (California or Florida). The state, action, and reward in equations (1), (2), and (3), respectively, are created by aggregating across jurisdictions in that State (CA or FL). However, the environment retain the mixing across jurisdictions. The agent is identifying an optimal policy for the full state.

I-SARL: In independent-SARL, each jurisdiction is an independent epidemic environment, i.e., there is no jurisdictional partnership mixing. Each agent is evaluating an optimal policy for its own jurisdiction. As there is no epidemic interactions, the actions of other agents does not impact its state transitions.

The specific jurisdictions used in each state are detailed below.

**California**: Alameda County (CA1), Los Angeles County (CA2), Orange County CA (CA3), Riverside County (CA4), Sacramento County (CA5), San Bernardino County (CA6), San Diego County (CA7), (rest of) California (CA8)

**Florida**: Broward County (FL1), Duval County (FL2), Hillsborough County (FL3), Miami-Dade County (FL4), Orange County FL (FL5), Palm Beach County (FL6), Pinellas County (FL7), (rest of) Florida (FL8)

The values of the parameters in Algorithm 1 were set as follows: $T = 12$, $M = 100,000$ and the optimizer was Adam. Figures A.2, A.3, A.4 and A.5 in the Appendix show the convergence in learning in MARL and SARL in California and Florida, respectively.

The budget allocation for each jurisdiction used in the experiment was obtained from Health Resource and Service Administration website. Ryan White HIV/AIDS Program (RWHAP) has allocated funds as EHE awards to 49 recipients that includes 39 metropolitan areas and 8 states HRSA. The total budget allocated to two states of interest, California and Florida was about $54.3 million and $48.4 million, respectively. While this budget from RWHAP has been used for our numerical study, it reflects only a fraction of the overall resources spent for HIV, and nationally, about 50% of people diagnosed with HIV have received some type of service from RWHAP. The total HIV related cost, which includes testing cost, retention-in-care cost and PrEP cost, was estimated from the simulator. The testing cost and retention-in-care cost were calculated using the cost functions defined in Khatami & Gopalappa (2021), whereas the annual PrEP related costs were taken from Sansom et al. (2021).

The feasibility of implementing an action was analyzed to set a constraint on the maximum possible action choices. Khatami & Gopalappa (2021) looked at the past trend in HIV proportion to set the feasible actions, specifically from 2010 to 2014, and found 2.3 % decrease in unaware proportion and 13.8 % increase in proportion of PWH in ART. Looking at the similar trend from 2015 to 2019, we found 1.6 % decrease in unaware proportion (14.9 % in 2015 to 13.3 % in 2019) CDC (a) and an increase of 4.8 % in VLS proportion (59.8 % in 2015 to 64.6 % in 2019) CDC (b) during that 5 years time interval. Thus, we set the feasible range of action choices as 0-0.5 % decrease in unaware proportion and 0-4 % increase in ART proportion each year. Additionally, the percentage of people with PrEP indicators that have been prescribed PrEP has gone from 13.2 % in 2017 to 30.1 % in 2021 which marks an increase of 16.9 % during that 5 years AHEAD (b). Thus, the third action, increase in PrEP proportion, was given a range of 0-4 % each year.

## 4.2 RESULTS

Figure 2 a) and b) depict the total new incidence in California and Florida from 2019 to 2030 MARL and A-SARL. In both states, MARL significantly outperforms A-SARL. In California, there is a notable 19% reduction in incidence when using the MARL approach, while A-SARL results in a 4.4% increase in total incidence from 2019 to 2030. A similar trend is observed for Florida, where MARL yields a 23% decrease in new incidence, while A-SARL results in a 0.6% increases. When comparing these incidence numbers for each jurisdiction within these states (results are available in the Appendix), a consistent pattern emerges in which the MARL approach outperforms the A-SARL approach across all jurisdictions. Figure 2 c) and d) display the differential costs incurred in California and Florida between MARL and A-SARL. As larger jurisdictions have higher prevalence, the decisions are biased towards larger jurisdictions resulting in higher intervention rates even for jurisdictions with lower prevalence as well. Thus, it adds to the cost of the interventions, without much impact in incidence in the lower prevalence jurisdictions. Although the bias in policies here is towards larger jurisdictions, this may not always be true such as when larger jurisdictions don't have a higher prevalence. Nonetheless, the results highlight the advantage of multi-jurisdictional modeling compared to a single agent national model. Results from MARL and I-SARL generate different optimal policies for the same jurisdiction, Figure A.1 a) shows an example for one jurisdiction. Without loss of generality we can conclude that excluding jurisdictional interactions (I-SARL) would generate different results than in MARL, highlighting the significance of jurisdictional mixing.

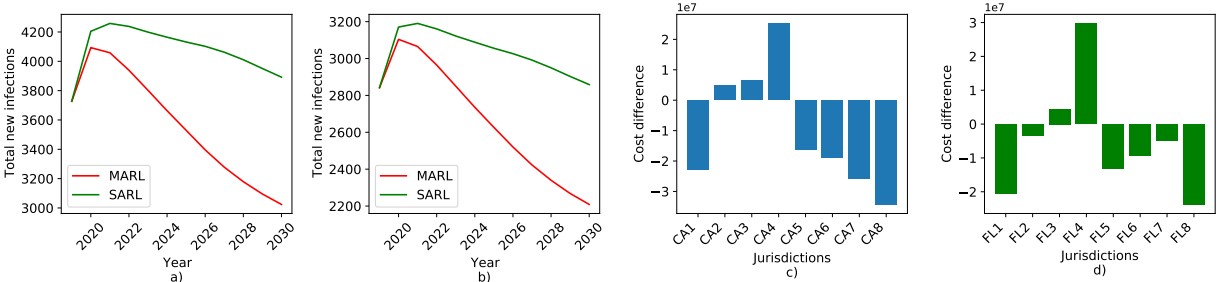

Figure 2: Comparing total new incidence from A-SARL and MARL in a) California and b) Florida, and cost difference (MARL minus A-SARL) in c) California and d) Florida.

Despite the superior performance of MARL, it is still a considerable distance from achieving the EHE targets. To move closer to the ambitious EHE initiative goal, a more aggressive policy approach is needed, involving increased intervention measures and a greater allocation of resources. To assess the effectiveness of these approaches in reducing HIV incidence, we modeled two different scenarios. The first scenario involves doubling the range of the action space, giving the agent more freedom to choose actions that result in higher levels of intervention. Specifically, we increased the action range to 1% decrease in unaware proportion, 8% increase in ART proportions and 8% increased in PrEP proportions. Figure 3 a) and c) illustrate that expanding the action range leads to a more significant reduction in the number of incidences by 2030, amounting to approximately a 34% reduction in California and 35% reduction in Florida. The second scenario involves increasing the budget allocation while maintaining the expanded action ranges. Figure 3 b) and d) illustrate the comparison of MARL in California and Florida with the initial budget and the increased budget, which is ten times the initial allocation. The new incidence trajectory demonstrates a reduction of approximately 58% from 2019 to 2030 in California while showing about 75% reduction in Florida in the same time interval. Although this level of reduction still falls short of the EHE goal, these scenarios clearly outline a viable pathway towards achieving those targets. The key takeaway is the need to allocate more resources and adopt more aggressive intervention measures in the pursuit of HIV elimination.

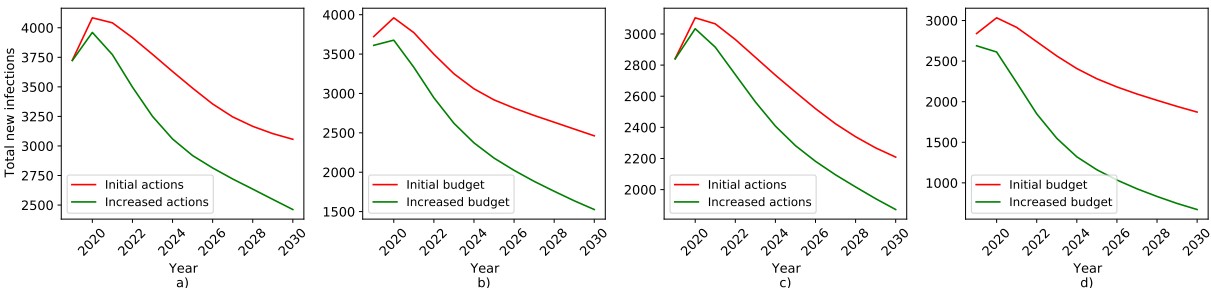

Figure 3: Comparing total new incidence from MARL with a) initial & increased actions and b) initial & increased budget in California, and c) initial & increased actions and d) initial & increased budget in Florida.

## 5 CONCLUSIONS

This paper explored a MARL decision-analytic framework for evaluating optimal intervention measures, (combinations of testing, treatment, and prevention) for ending the HIV epidemic. The MARL framework empowers each jurisdiction to make independent decisions tailored to its specific context, in the context of epidemic interactions across jurisdictions. MARL was compared with A-SARL and I-SARL, which are representative of current decision-analytic frameworks in the HIV modeling literature. Differences in results suggest that national aggregated analyses (A-SARL) or independent (no interactions) jurisdictional analyses (I-SARL) can lead to sub-optimal decisions.

## REPRODUCIBILITY STATEMENT

The following steps help with the reproducibility of the results presented in this paper: 1) The model framework can be implemented with Algorithm 1 proposed in Section **??**. 2) The environment for RL (defined as simulation model in 3.1) is taken from Tatapudi & Gopalappa (2022) which works as the simulator for our model. 3) The hyperparameters used in model training are given in Table A1. 4) The source code and data sets are made available for implementation via the anonymous GitHub link.

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

# A   APPENDIX

## A.1   TERMINOLOGY DEFINITION

**Ending the HIV Epidemic (EHE)** - Initiative by U.S. Department of Health and Health Service (HHS) to reduce new HIV infections in U.S. by 75% by 2025 and by 90% by 2030

**Jurisdiction** - Counties and states that contribute significantly to total HIV infections and are used in the HIV modeling

**HIV prevalence** - Number of persons living with HIV at a given time regardless of their time of infection, whether the person has received the diagnosis, or the stage of HIV disease

**Antiretroviral Therapy (ART)** - Treatment of HIV that involves taking a combination of HIV medicines everyday

**Viral Load Suppression (VLS)** - Having less than 200 copies of HIV per milliliter of blood

**Pre-exposure Prophylaxis (PrEP)** - Medicine taken by people at risk for HIV to prevent getting HIV from sex or injection drug use

**HIV care continuum** - Steps that people with HIV take from diagnosis to achieving and maintaining viral load suppression

**Retention-in-care** - Adherence to drug regimens and other clinical or lifestyle-change recommendations made by the providers

**Compartmental Simulation Model** - Model with various compartments where each individual in a population can be on one compartment and total population is the sum of all the b compartments

**CD4 Count** - Blood test that measures the number of CD4 cells in the blood

**Heterosexual Male/Female** - People who have heterosexual contact

**Men who have sex with men (MSM)** - People who have male to male sexual contact and men who have sexual contact with both men and women

**Diagnostic rate** - Estimated number of new HIV diagnosis among the people unaware of their HIV infection

**Dropout rate** - Estimated number of people leaving the HIV treatment regime among the people in HIV care

## A.2 HYPERPARAMETERS

Table A1: List of hyperparameters used in PPO

| Hyperparameters | Value |
|---|---|
| Discount factor | 0.99 |
| Actor learning rate | 0.0003 |
| Critic learning rate | 0.0003 |
| Initial exploration | 0.4 |
| Final exploration | 0.05 |
| Decay rate | 0.0046 |
| Decay frequency | 1000 |
| PPO clip parameter | 0.2 |
| K epochs | 20 |

## A.3 MIXING ASSUMPTIONS

Figure A.1 a) shows the incidence estimates considering mixing assumptions compared to the scenario without mixing where each jurisdiction is trained on its own. As the network is trained in the single agent and tested with mixing considered among all the jurisdictions, we can see individual jurisdictions model underestimates the total incidence numbers. Therefore, mixing assumptions across the interacting jurisdictions should be considered to get a better estimates of incidence in any jurisdiction. This also justifies the need of a combined modeling as a single jurisdiction (or city) model might not be able to capture the proper estimates of new infections.

Table A2: Sexual partnership mixing among jurisdictions Tatapudi & Gopalappa (2022)

| Jurisdiction interaction | HM | HF | MSM |
|---|---|---|---|
| Same jurisdiction | 57% | 65% | 47% |
| Other jurisdiction in same state | 28% | 23% | 31% |
| Other states | 14% | 12% | 22% |

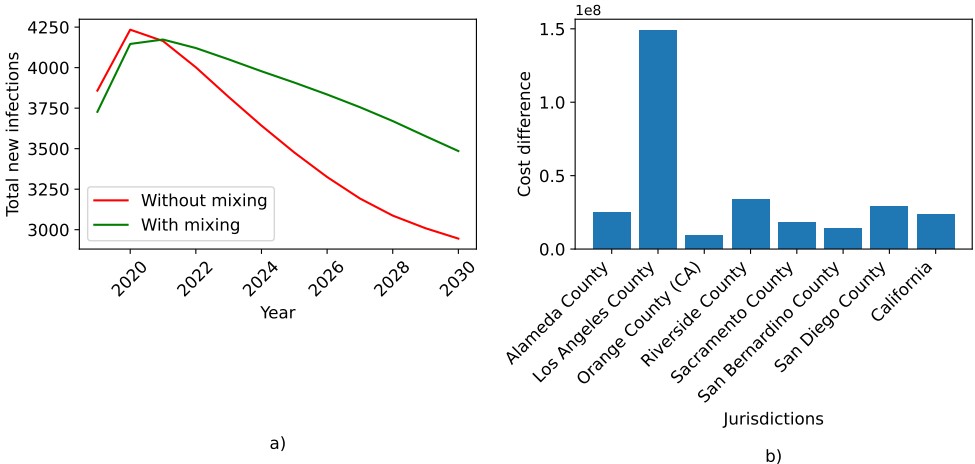

a)

b)

Figure A.1: Comparing a) new incidence with mixing (MARL) and without mixing (I-SARL) and b) cost difference (without mixing - with mixing) in California.

## A.4 REWARD

Figures A.2 and A.4 show the learning curve for MARL and A-SARL approaches respectively for jurisdictions in California. Figures A.3 and A.5 show the learning curve for MARL and A-SARL approaches respectively for jurisdictions in Florida.

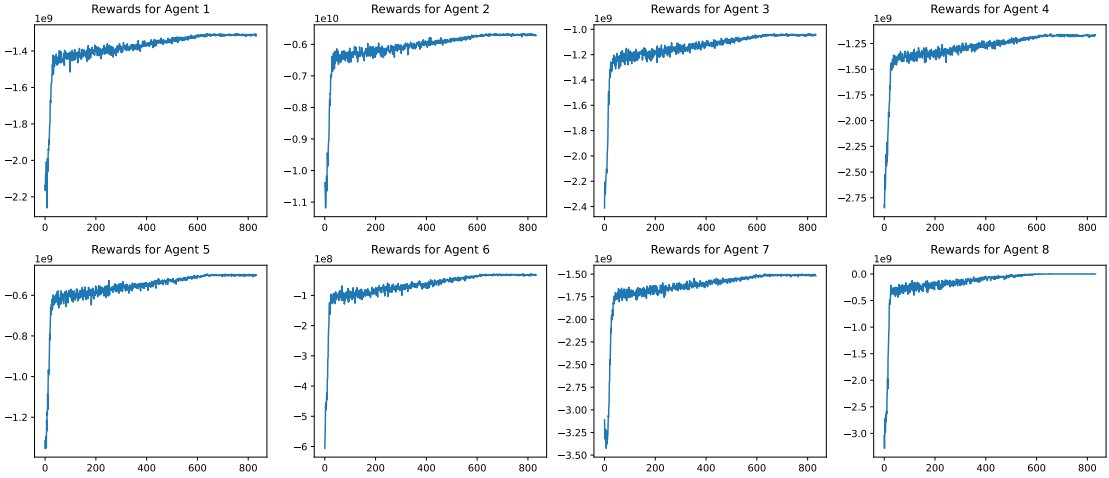

Figure A.2: Reward curve for MARL agents in California.

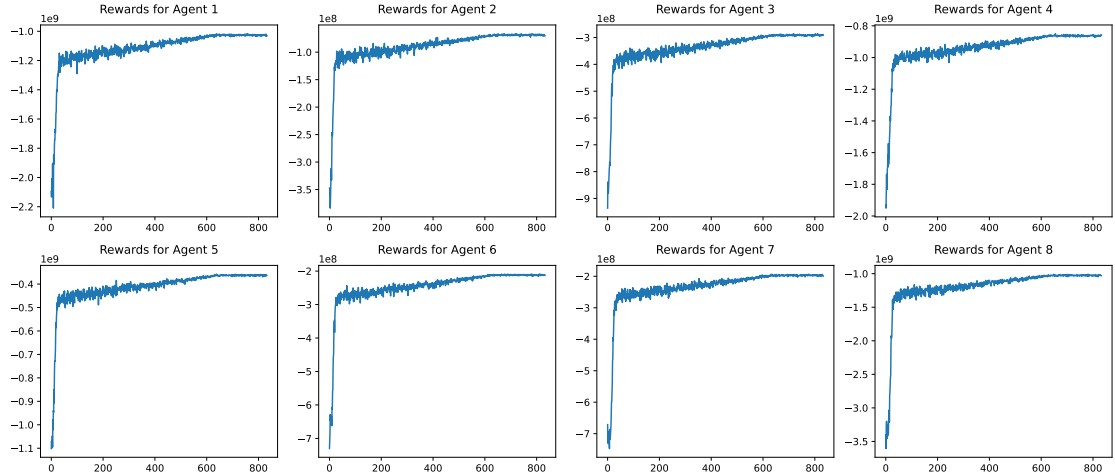

Figure A.3: Reward curve for MARL agents in Florida.

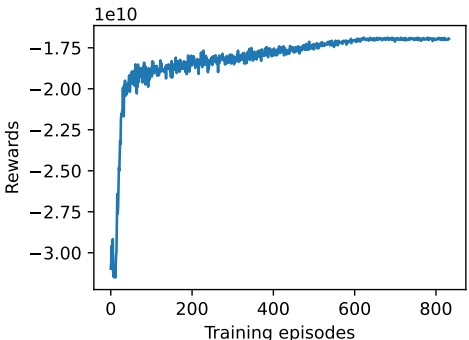

Figure A.4: Reward curve for A-SARL agent in California.

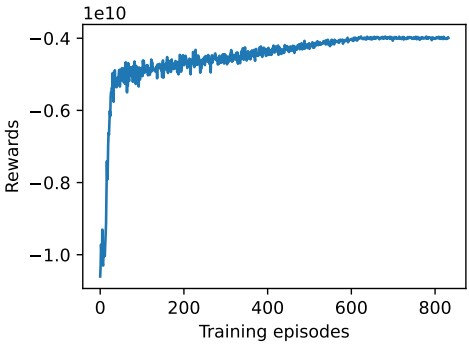

Figure A.5: Reward curve for A-SARL agent in Florida.

## A.5 NEW INCIDENCE

Below is the overall incidence number for all the jurisdictions in each of the states, California and Florida.

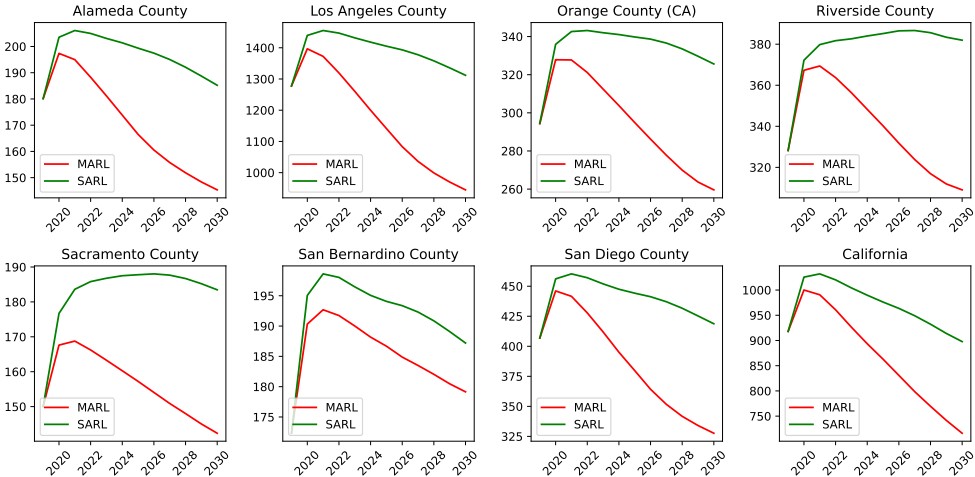

Figure A.6: Comparing new incidence with A-SARL and MARL in California.

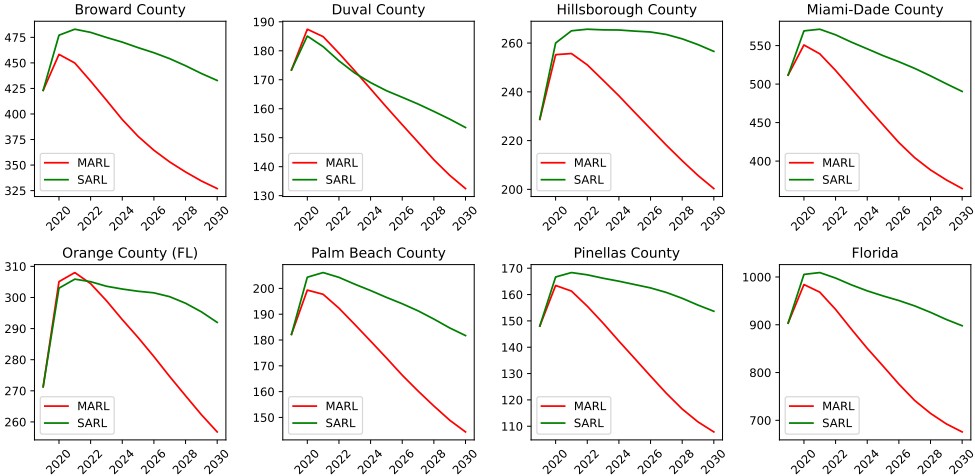

Figure A.7: Comparing new incidence with A-SARL and MARL in Florida.

