# OpenReview forum: "A Multi-Agent Reinforcement Learning Framework for Evaluating the U.S. ‘Ending the HIV Epidemic’ initiative"
_ICLR.cc/2024/Conference — Submitted to ICLR 2024_

### Official Review · Reviewer_N95H · 2023-11-01

**Soundness:** 2 fair
**Presentation:** 1 poor
**Contribution:** 2 fair
**Rating:** 5
**Confidence:** 3

**Summary:**

This paper aims to achieve the EHE initiative by applying MARL to explore optimal combinations of actions at the jurisdictional level while considering cross-jurisdictional epidemiological interactions. By using the compartmental simulation model and training multiple agents based on PPO, this paper shows the effectiveness of MARL over SARL.

**Strengths:**

This paper uses recent machine learning tools to solve an important public health challenge of HIV.

**Weaknesses:**

1. In general, the paper's current writing has many public health terms. Thus, without much prior knowledge of HIV and public health terms, the paper is difficult to read.
2. It is unclear whether a comparable single-agent baseline is used. According to Section 3.3.2, SARL formulation applies $j=1$, so SARL outputs one action every timestep whereas MARL outputs $N$ actions every timestep. If this is true, then MARL outperforming the SARL baseline could be a straightforward result (due to outputting more actions). Instead, a more competitive baseline could be a centralized agent that outputs $N$ actions instead of one action.
3. SOTA MARL applies centralized training and decentralized critic with a centralized critic. However, Section 3.3 applies multiple single-agent algorithms (i.e., multiple PPO) without the use of centralized critics.
4. Results in Figures 2 and 3 need multiple seeds for statistical significance.
5. Because multiple agents are interacting in the environment, MDP (Section 2) may not be the correct term, and a Markov game (Littman 1994) would be a more correct term to use.

Michael L. Littman. Markov games as a framework for multi-agent reinforcement learning. 1994

**Questions:**

I hope to ask the authors' responses to my concerns (please refer to the weaknesses section for details).

---

> ### Author Response · Authors · 2023-11-23
>
> Thank you for the insightful review.
>
> 1.	Thank you very much for raising this concern. We understand that the paper might be difficult to comprehend for the readers outside of the HIV and public health domain. We have added a terminology definition in the Appendix of the paper.
>
> 2.	The reviewer is correct that the main difference between MARL and SARL is that MARL identifies an optimal policy tailored to a jurisdiction while SARL identifies an aggregated national optimal policy. We now refer to SARL as aggregated-SARL ot A-SARL. Kindly note that we compared MARL with two different models (now referred to as A-SARL and Independent-SARL or I-SARL), each representing the two broad categories in current HIV modeling literature. To note, current models in the literature are either national i.e., evaluate one aggregated policy for the nation, or independent jurisdictional, i.e., do not model the mixing across jurisdictions but estimate actions specific to that jurisdiction. We formulated A-SARL to represent the former. To represent models of the latter type, we modeled I-SARL, where we evaluated one jurisdiction independently, i.e., evaluated optimal policy under assumption of no cross-jurisdictional interactions (no-partnership-mixing). Figure A.1 (a) compares results between MARL and I-SARL. Results suggest that discarding the effect of mixing leads to different policies with different impacts. Without loss of generality, we can conclude that if we do this analyses for each of the N jurisdictions, i.e., estimate optimal policy for each of the N independent jurisdictions, the resulting optimal policy will differ from that obtained from MARL. We believe the reviewer’s suggestion on having a centralized agent that output N actions can be considered representative of this type of analyses.
> We have rewritten the introduction and modified several parts of the paper to more clearly discuss this.
>
> 3.
> This is a great suggestion. We are trying to use a similar method for the extension of our work over all the jurisdiction in the US. In this paper, our aim was to evaluate the advantages of multi-agent approach compared to single agent, and thus, we used a simple case of multi-agent reinforcement learning.
>
> 4.
> We used multiple seeds in our training to show the convergence of the algorithm. Due to page limitations, we did not include those results from various random seeds. We will add a short description of this in the Appendix.
>
> 5.
> Though there are interactions across agents we assume a non-competitive and cooperative environment, as is the case in reality. Though Markov games is a broader term that can include these variations, we use the term MDP to avoid implications of competition between jurisdictions.

---

### Official Review · Reviewer_uaHW · 2023-11-01

**Soundness:** 2 fair
**Presentation:** 3 good
**Contribution:** 2 fair
**Rating:** 5
**Confidence:** 2

**Summary:**

The authors propose a multi-agent model for modeling interventions by different entities in the US to end the HIV epidemic in the US. They find that using MARL in this multi-agent framework yields better policies than using single agent RL and the associated single agent model. As part of this paper, they also provide a concrete multi-agent environment definition for HIV spread and control via interventions. In their environment each geographical jurisdiction is considered as a separate agent to model decentralized decision making.

**Strengths:**

The authors present a practical use case for deep MARL for a problem that potentially has an important social impact. The paper is well written and the concepts are clearly explained.

**Weaknesses:**

While the application described in this paper is for real-world use, it is not clear to me how useful this is as the state and more importantly the action spaces are considered to be very small. Also no validation is provided, which is important since this can affect real-world decision making.

**Questions:**

1. Is the difference between the single agent model and the multi-agent model only one of centralized versus decentralized decision making?
2. Are there situations where the rewards of one agent (jurisdiction) depends on the actions of other jurisdictions?
3. Is this multi-agent environment a fully cooperative one, in which case, this can be modelled as a Dec-POMDP.
4. Since the algorithm is run only for $T$ timesteps, shouldn't the author use a finite horizon MDP framework, where the timestep should also be added to the state?
5. Are there any resource constraints across all jurisdictions that need to be modeled?
6. Is the difference in the performance of the SARL and MARL policies a result of sub-optimality in learning or is it because of some other structural difference between the single and multi-agent environment?
7. How can the solutions be validated? Is there any validation for the environment? What are other precautions to be taken before using these solutions in the real world?
8. In such an environment, how does a baseline policy perform? Baseline policy could be the bestb policy from literature or if nothing is available for this environment, then a random policy could be used.

**Details Of Ethics Concerns:**

I think the authors should add a statement regarding the potential ethical implications of their work. Since this would be applied in the real-world for healthcare, any error in the algorithm could have significant real-world impact. The authors would also need to consider the effect on societal impact and fairness of their proposed solution.

---

> ### Author Response · Authors · 2023-11-23
>
> Thank you for the insightful questions.
>
> 1.
> Yes, that is correct. In the single-agent (now aggregated-SARL or A-SARL), one aggregated national-level action is selected and applied to all jurisdictions. In the multi-agent model (MARL), each agent is evaluating an action specific to its own jurisdiction, which makes it decentralized. We had a third scenario that was presented in the appendix, which we now refer to as independent-SARL or I-SARL, which represents evaluating each jurisdiction independently, i.e., ignoring the epidemic interactions across jurisdictions. A-SARL and I-SARL are representative of models that are used in current HIV literature.
> We have rewritten the introduction and modified several parts of the paper to more clearly discuss this.
>
> 2.
> Yes. The agents are acting in an environment with epidemic interactions across jurisdictions, i.e., sexual partnerships can form between persons of different jurisdictions. Thus, the actions taken in one jurisdiction will affect the infection rate of its neighboring jurisdictions, i.e., the state transitions and rewards of each jurisdiction are influenced by the actions taken by all jurisdictions. We made modifications to the methods to clarify these points.
> 3.
> We assume it is fully cooperative as  in reality. In the model, while the current state is known to all agents and agents share a common objective, as each agent evaluates its own action and all agents simultaneously evaluate their actions, the evaluated state-transition will be different than actual state transition. Therefore, from an agent’s perspective, the environment presents properties of non-stationarity, making the formulation equivalent to Dec-POMDP in model-based methods. For purposes of scalability or in the model-free context as ours, MARL methods have been evaluated as approximate solution algorithms for Dec-POMDP [Hyun-Rok et. al. Operations Research, 2021; Shayegan, et al., PMLR, 2017].
> In this work, we focused on evaluating the significance of MARL instead of SARL as in current HV modeling literature. In future work, in application of the model to the U.S, we plan on evaluating multiple solution algorithms suitable for this context.
> 4.
> That is a great point that, generally, this application would require a non-stationary policy if modeling epidemic prevalence as the state, and including time-step ‘t’ into the state space can do that. In our case, though we did not prove this and is a limitation, the chance that the same state occurs at two different time-points is low, because of a few points related to our formulation and the application features as follows. Our state is formulated as a conjunction of prevalence (proportion infected) and care metrics (proportion aware, proportion VLS, and proportion PrEP). We set constraints on care metrics to be non-decreasing over-time. HIV has no cure, and therefore unless deaths exceed incidence, prevalence is non-decreasing. In the case that deaths are lower than incidence, i.e., prevalence is non-decreasing over time: If care metrics increases at a slower rate, prevalence increases at a faster rate, and thus, having the same care metrics at two time-points would have two difference prevalence, and similarly, having same prevalence at two time-points can only be generated from two different care metrics. In the case that deaths are lower than incidence, prevalence can decrease over time. However, as people with HIV can live almost normal lives, deaths can exceed incidence only when incidence is very low, which can happen only if care metrics are very high. Therefore, as before, the same prevalence cannot be generated by two different care metrics and vice-versa. However, without a formal proof, there is a chance that we may overlook certain scenarios that may lead to same state at two different time-points  (a limitation of our work), but we believe the chance of those scenarios are small. Adding a ‘t’ would increase complexity of training, so our current formulation is more desirable. Though adding care metrics into the state formulation also increased the complexity, it has several other useful properties (as shown in previous work [1]). However, with the availability of more time, we hope to conduct numerical analyses that include ‘t’ in the state-space to evaluate its significance in our current application.
>
> 5.
> Although every jurisdiction has different sources of funding, public and private, in economic modeling terms, we use a ‘societal perspective’, as is the typical case when evaluating national policies. Specifically, the national policy sets the guidelines, then the necessary resources are allocated through multiple funding sources.
> Here we assumed a fixed national budget, and evaluated how to optimally allocate it across jurisdictions. In application of the model to inform the EHE (Ending the HIV Epidemic), we will evaluate optimal policies under different budgets, to collective identify resource needs and optimal policy.

---

> ### Author Response · Authors · 2023-11-23
>
> 6.
> The environments are similar in both SARL (now  A-SARL) and MARL. The difference is that MARL identifies an optimal policy for ‘each’ jurisdiction while A-SARL identifies an ‘aggregated’ national optimal policy. Kindly note that we compared MARL with two different models (now A-SARL and I-SARL), each representing the two broad categories in current HIV modeling literature. Kindly note that the scope of this paper was to demonstrate the use of MARL to overcome current gaps in the HIV modeling literature.
> We have rewritten the introduction and modified parts of the paper to clarify this.
>
> 7.
> We agree that it is extremely important to have validation steps. Thus far we have taken precautions to extensively validate the environment (in prior work Tatapudi et. al., 2022]) and in data inputs. The data inputs are from well-established CDC’s survey and surveillance systems, with detailed cost functions informed through the literature. However, the data for the cost functions are limited when related to outreach and support programs, or as new technologies evolve (e.g., higher efficacy in HIV self-tests, which may have more uptake than clinic-based tests). Therefore, in future work, when applying this model to inform policy, we plan to do extensive sensitivity analyses. We also additionally plan to compare our model results with results from status-quo policy, under the range of uncertainty assumptions for all the sensitive input values, and under varying cost functions. This will help to evaluate if our model’s optimal policy would outperform the status-quo policy in all input assumptions or only in certain cases, which will help make informed decisions.
>
> 8.
> The status quo policy acts as the baseline policy in our problem.

---

### Official Review · Reviewer_Trsi · 2023-11-01

**Soundness:** 1 poor
**Presentation:** 2 fair
**Contribution:** 2 fair
**Rating:** 3
**Confidence:** 4

**Summary:**

This paper proposes a reinforcement learning-based approach to evaluate the effectiveness of several intervention measures to minimise new HIV infections. Throughout the paper the authors formalize the HIV epidemic environment following the principles of RL and show how the framework can be used to tackle new infections.

**Strengths:**

The paper evaluates the impact of the HIV epidemic within US jurisdictions from a reinforcement learning perspective. This can be good since machine learning methods that model these scenarios can be leveraged to minimise the impact of these serious health concerns. The authors also analyse the impact of specific mitigation measures within their approach in detail, and look at the problem from a multi-agent perspective, which is important since different areas of impact may have different requirements and conditions.

**Weaknesses:**

Overall, this paper contains several inaccuracies and technical flaws from a MARL perspective. I outline here some points and more questions ahead.

* In section 1 the authors state that "To our knowledge, no prior research has explored dynamic jurisdictional interactions in these models."; while I accept this claim for HIV cases, it is important to note that governmental actions and interventions have been included in cases related to the study of pandemics or epidemics such as COVID-19 [1] or influenza [2, 3]. These are modelled using SIR and SEIR models that are not mentioned in this paper but can also be relevant when it comes to epidemic models.
* In section 3.2: "The MDP can be defined as a tuple"; The MDP reference is missing; also, it is not correct to define an MDP as a system with multiple agents; an MDP is formed by a single agent only, and it can indeed be extended to multi-agent MDPs [4].
* In section 3.2 it is described that each agent has a different individual observation but figure 1 gives the idea that all of them share the same state since these are represented as $S_t^j$; additionally, if the authors are considering individual observations in their MARL model, the problem should instead be modelled as a Dec-POMDP [5]; an MDP considers a fully observable state.
* In section 3.2 "For each agent j, $a^j$ signifies their set of possible actions": $a^j$ is defined here as a set but ahead we can see that it denotes the action of agent $j$: $a = a^1 \times ... a^N$. I have outlined a few inaccuracies but, generally, the entire section 3.2 needs to be revised since it contains multiple inaccuracies.

Minor comments:
* "The MDP can be defined as a tuple" in section 3.2: missing brackets in the tuple
* Missing brackets in many in-text citations
* Missing full stop at the end of some equations such as (4) or (7)
* Before equation (5) the full stop in "with the following objective." is misplaced

Generally, while the problem being analysed is interesting, I have several questions regarding the MARL formulations.

[1] https://www.sciencedirect.com/science/article/pii/S120197122030117X

[2] https://www.ncbi.nlm.nih.gov/pmc/articles/PMC5853779

[3] https://ceur-ws.org/Vol-2563/aics_19.pdf

[4] https://dl.acm.org/doi/pdf/10.5555/1029693.1029710

[5] https://www.fransoliehoek.net/docs/OliehoekAmato16book.pdf

**Questions:**

1. In section 3.2.1 the proportion of PWH unaware of the infection is used as part of the state; is it possible in real life to know this proportion if people are not aware that they are infected? Does it affect the model if this number is unknown?
2. According to Algorithm 1, each episode is T years long. This means that each time step in an episode is one year; this makes me question whether this is a reasonable choice, since it means that we take only one action per year. Is that enough? during a year many changes might happen and several governmental measures might be needed.
3. It is unclear to me how MARL is being used here when compared to the SARL approach. From my understanding, in the SARL approach, there is a state (as per eq (1)) that describes the conditions of all the jurisdictions within a state (using the average, as stated in section 3.3.2), whereas in the MARL approach we have the same state but each jurisdiction has its own independent values; then, in SARL the actions performed influence the values in the state for all jurisdictions and then in MARL each jurisdiction performs an action that influences only its own state conditions. How are the multi-agent interactions between different jurisdictions integrated here? In order to say that this is following a MARL approach there needs to be some sort of interaction among the agents that will make them cooperate or work together towards some common goal. In the presented approach, I do not see that being done. Even the rewards are said to be given individually and its components only correspond to jurisdiction $j$ (as per eq (3)). This means that each agent is maximising an independent objective on its own, without any effect on the other agents. This sounds like multiple single-agent problems happening in parallel, without any interaction.

---

> ### Author Response · Authors · 2023-11-23
>
> Thank you for the insightful review.
>
> 1.	That is correct that for infectious diseases, typically in real-life, proportion unaware are not always know. However, for HIV, the surveillance methods are well advanced that there exists good molecular methods that have been shown to provide accurate estimates. They are routinely estimated as part of CDC’s NHSS (National HIV/AIDS Surveillance Systems) and made publicly available annually.
> Previous work [Khatami, 2021] showed that this representation of state with the action representation of directly using changes in proportion unaware (instead of testing frequency, which is more implementable) has very good mathematical properties suitable for convergence. First, it removes the infeasible solutions, and second it makes the feasible space convex, thus assisting in faster convergence. Further, once we determine the optimal action (i.e., proportion change in unaware) they can be converted into testing frequency, i.e., given a state, a function mapping the action to testing frequency is a bijection function (one-to-one mapping).
>
> 2.	That is correct that some infections, such as the COVID-19 pandemic, required constant policy changes to quickly adapt to the changing pandemic given: a) COVID-19 is a fast spreading disease (transmission rate per contact per day ~14%), b) it was a newly emerging infectious disease with no prior knowledge for setting appropriate intervention guidelines, and thus, frequent changes were needed, and c) we believe, it was feasible to implement frequent changes under the declaration of a global emergency.
> On the contrary, HIV is a slow spreading disease (transmission rate of 4% per year), and thus, the impact of frequent changes would not be feasible to evaluate as part of routine monitoring. The national strategic plans and guidelines are modified less frequently, e.g., the most recent changes in HIV testing guidelines were in 2017 and only for one population (MSM) [2]. Further, we believe, frequent changes in public health guidelines may not be feasible in the case of non-emergency situations given the time it takes to implement changes, such as US congressional appropriations, preparation of healthcare professionals and infrastructure, and changes in insurance policy.
> However, we chose a shorter timeline of 1 year to provide the flexibility to balance feasibility with need . For example, PrEP guidelines were updated in 2022 because a new drug was FDA approved in 2021. Estimating optimal annual rates provides the flexibility to set appropriate guidelines. For example, with recent advances in HIV take-home self-test kits, there is interest in suitable guidelines for combinations of clinic-based tests v. self-tests (weighted by effectiveness, adherence, etc.) to reach diagnostic rate targets to achieve the EHE (Ending the HIV Epidemic) goal. Our approach of identifying optimal annual diagnostic rate targets fits well in this context.
>
> 3. That is correct that in the case of SARL, a single agent learns the actions for all the jurisdictions in it. But in the case of MARL, each jurisdiction (agent) learns its own actions. The interactions across jurisdictions in both MARL and SARL are in the epidemic environment, modeled as sexual partnership mixing across jurisdictions.
> In the setup of our MARL model, each agent is determining an optimal action based on the current state of the system (current state of each jurisdiction is known to all jurisdictions). However, as state transitions depend on the actions implemented in all jurisdictions (because of sexual partnership mixing), and as all agents (jurisdictions) are simultaneously evaluating this decision, it is infeasible to know the actual state transition. This introduces properties of non-stationarity into state-transitions, making this a suitable application for MARL.
> We also evaluated a SARL (now called aggregated-SARL or A-SARL) formulation as a representation of current models in HIV literature. Specifically, the models in the HIV literature are typically ‘aggregated’ national or ‘independent’ jurisdictional (no epidemic mixing across jurisdictions). We use A-SARL as a representation of the former type. To represent models of the latter type, we evaluated one jurisdiction independently, i.e., ignored the mixing (presented in Appendix Figure A1). We now refer to this as independent-SARL or I-SARL. The corresponding results differed from that of MARL highlighting that ignoring mixing can lead to non-optimal decisions.
> The scope of this work was to present the use of MARL to overcome the gaps in current HIV modeling literature. We believe our contribution is developing a comprehensive modeling framework and simulation environment that is suitable for analyses of the U.S. ‘Ending the HIV Epidemic’ national strategic plan.
> We have rewritten the introduction and modified several parts of the paper to more clearly discuss this.

---

### Official Review · Reviewer_TfZh · 2023-11-03

**Soundness:** 3 good
**Presentation:** 3 good
**Contribution:** 2 fair
**Rating:** 5
**Confidence:** 2

**Summary:**

This paper aims to simulate a real-world optimization problem and propose to solve it under MARL setting due the factored-controller nature of the problem.

**Strengths:**

Well written background and related work. Clear on the method used (PPO) and the simulation environment. The paper aim to solve a real-world problem, which should be considered a positivity due to its applicable nature, but I'm not entire sure this holds for ICLR.

**Weaknesses:**

The novelty of the proposed approach is limited, the problem can be viewed as a joint-action problem, solving it as a MMDP, hence viewing it as a factored action problem, does not change the problem and the learning process fundamentally. The simulation itself is also not novel to this paper, and only method (PPO) is used for evaluation.

**Questions:**

What would be the theory or explanation of the increased performance and why not factor the actions in other ways (or conjunction of jurisdiction) such as separating the action space further into different controllers for testing, treatment and prevention?

---

> ### Author Response · Authors · 2023-11-23
>
> Thank you for the insightful review. The main difference between the multi-agent model (MARL) and single-agent model (SARL) is that MARL identifies an optimal policy tailored to a jurisdiction while SARL identifies an aggregated optimal policy. The scope of this work was to evaluate the advantage of a MARL type model, as opposed to current modeling approaches in the literature. Specifically, current models in the literature are either national, i.e., do not model jurisdictional heterogeneity but instead evaluate aggregated national-level decisions, or independent jurisdictional, i.e., do not model the interactions (through sexual partnership mixing) across jurisdictions but estimate actions specific to that jurisdiction. We formulated SARL (which we no refer to as aggregated-SARL or A-SARL) to represent the former, for purposes of comparing with our proposed MARL approach. To also note, to represent models of the latter type, we evaluated one jurisdiction independently, i.e., evaluated optimal policy under assumption of no interactions (no partnership-mixing) with other jurisdictions and compared it to results from MARL for that jurisdiction. We now refer to this as independent-SARL or I-SARL. We presented these analyses in the Appendix, in Figure A.1 (a), and the observations are that discarding the effect of mixing leads to different policies with different impacts.
> The reviewer makes a good point that we could have expanded the action and state space to include a conjunction of all jurisdictions, which would then be reduced to a single-agent model. However, such a model is not scalable. Though in these analyses we used a simple example of 8 jurisdictions, in reality the number of jurisdictions are typically large (96 in our ongoing work applying this model to the national U.S). This creates massive sizes of state and action space. In this example, the size of the action space would be 9^96, which makes convergence challenging. Therefore, we believe, MARL is a suitable approach for this problem
> To note that, the scope of this work was to demonstrate the advantage of MARL to overcome the gaps in current HIV modeling literature. We believe our contribution is developing a comprehensive modeling framework and simulation environment that is suitable for analyses of the Ending the HIV Epidemic national strategic plan.
> We have rewritten the introduction and modified several parts of the paper to more clearly discuss this.

---

### Meta-Review · Area_Chair_Tfoh · 2023-12-09

**Metareview:**

a) Claims: The paper compares the use of multi-agent RL to that of single-agent RL to plan interventions for reducing new HIV infections.  MARL produces qualitatively different strategies than SARL.

b) Strengths: All reviewers agreed that the paper uses ML to address an important real-world public health problem.

c) Weaknesses: The reviewers all found this paper to be below the bar for publication.  Multiple reviewers identified several technical imprecisions.  There was a consensus that the underlying problem is not multi-agent; rather, a single agent needs to learn a factored action that might choose different interventions in different jurisdictions.  The apparent different in results seems to stem from the SARL approach treating all jurisdictions the same.

One reviewer flagged the need for an ethics statement.  Given the high-stakes nature of the decisions being studied, I strongly encourage the authors to add a consideration of disparate impacts and fairness to future revisions.

**Justification For Why Not Higher Score:**

The scientific content is the comparison between SARL and MARL approaches; however, I am convinced by the reviews that the wrong SARL approach is used as a baseline.

**Justification For Why Not Lower Score:**

n/a

---

### Decision · Program_Chairs · 2024-01-16

Reject